# Improved CNN-Based Indoor Localization by Using RGB Images and DBSCAN Algorithm

**DOI:** 10.3390/s22239531

**Published:** 2022-12-06

**Authors:** Fang Cheng, Guofeng Niu, Zhizhong Zhang, Chengjie Hou

**Affiliations:** 1School of Electronic and Information Engineering, Nanjing University of Information Science and Technology, Nanjing 210044, China; 2School of Communication and Information Engineering, Chongqing University of Posts and Telecommunications, Chongqing 400065, China

**Keywords:** indoor location, convolution neural network (CNN), DBSCAN, Wi-Fi fingerprints

## Abstract

With the intense deployment of wireless systems and the widespread use of intelligent equipment, the requirement for indoor positioning services is increasing, and Wi-Fi fingerprinting has emerged as the most often used approach to identifying indoor target users. The construction time of the Wi-Fi received signal strength (RSS) fingerprint database is short, but the positioning performance is unstable and susceptible to noise. Meanwhile, to strengthen indoor positioning precision, a fingerprints algorithm based on a convolution neural network (CNN) is often used. However, the number of reference points participating in the location estimation has a great influence on the positioning accuracy. There is no standard for the number of reference points involved in position estimation by traditional methods. For the above problems, the grayscale images corresponding to RSS and angle of arrival are fused into RGB images to improve stability. This paper presents a position estimation method based on the density-based spatial clustering of applications with noise (DBSCAN) algorithm, which can select appropriate reference points according to the situation. DBSCAN analyses the CNN output and can choose the number of reference points based on the situation. Finally, the position is approximated using the weighted k-nearest neighbors. The results show that the calculation error of our proposed method is at least 0.1–0.3 m less than that of the traditional method.

## 1. Introduction

As location-based applications and services are associated with our daily lives, they have gained increased interest [1]. Due to the complicated traits of the indoor surroundings, the effects of shadow fading, the intensity of personnel, and the external weather environment, outdoor range-based localization technology cannot deliver accurate inside positioning services [2]. However, indoor positioning has a very important application in the field of architecture. Building emergency management [3], smart plug load control [4], and smart HVAC controls [5] are just a few typical examples. Indoor positioning has become a research hotspot.

Indoor positioning technology can be roughly classified into two types, namely, range-free and range-based. Time of arrival (TOA) [6], time difference of arrival (TDOA) [7], angle of arrival (AOA) [8], and received signal strength (RSS) [9] are examples of traditional range-based approaches. Range-based positioning technology needs to maintain the accuracy of positioning under certain conditions. For example, AOA requires complex hardware and algorithm support, and TOA and TDOA require clock synchronization between the receiver and transmitter. Importantly, the range-based indoor positioning is very unstable because it is easily affected by non-line-of-sight (NLOS) factors [10]. However, the range-free methods can effectively improve the influences of the above NLOS factors. The range-free methods mainly include radio frequency identification devices (RFIDs) [11,12], Bluetooth [13,14,15], ZigBee [16], Ultra Wideband (UWB) [17], Wi-Fi [18], and other technologies. The RFID has low power consumption and a wide range, but it has low positioning accuracy. Similarly, Bluetooth has high throughput, commonality, and low power consumption, but it has low positioning accuracy. The ZigBee positioning environment is suitable for wireless sensor networks. However, for the majority of users’ devices, it is not accessible. Wi-Fi can solve the shortcomings of these technologies. Wi-Fi is widely used in daily lives of people, and it does not require additional hardware devices. Wi-Fi-based indoor location can be located through the offline phase of information fingerprint collection and the online phase fingerprint matching. The important thing is that Wi-Fi has higher positioning accuracy.

Regarding Wi-Fi-based indoor positioning systems, radar was the first fingerprint system developed by Microsoft [19]. The radar system is based on the location fingerprint generated by the Wi-Fi network. Then, M Youssef et al. proposed the Horus system to improve the positioning accuracy by using the k-nearest-neighbor (KNN) probability method [20]. On the other hand, Chen et al. proposed to build an AOA fingerprint positioning system and remove noise by filtering method, which tremendously improved the robustness of the system and effectively enhanced the positioning accuracy [21]. However, in a complex indoor environment, signal propagation is easily influenced by external factors; hence, the developed system framework is unstable. According to relevant studies, CNN can also be applied to indoor positioning systems to achieve a better positioning effect by matching the most suitable position in the way of image classification [22,23,24]. Sinha and Hwang proposed a CNN model to solve the instability of fingerprint databases built with RSS [25]. Shao et al. created a directed acyclic graph (DAG) CNN positioning system to address the drawbacks of Wi-Fi and geomagnetic signal instability [26]. Hsieh et al. used RSS and channel state information (CSI) as CNN data input [27]. Li et al. used AOA, TOA, and amplitude of CSI data sources to build RGB images to obtain higher positioning accuracy [28]. Sinha and Li et al. have achieved a good positioning effect by using the CNN method for indoor positioning. All the above researches fused one or even several data sources as the input of CNN model. Inspired by these studies, we propose using RSS and AOA to construct RGB images using CNN model data input [29,30,31]. The detailed procedure will be shown in Section 2 and Section 3.

In general, the output selection and final position determination of the CNN model are also issues of concern. The KNN algorithm is used to estimate the final position [32]. The traditional fingerprint database calculates the estimation of the final target position by setting a fixed value of K. The advantage of the traditional fingerprint database is simple and easy to implement, and it may cause large errors and decrease the positioning effect. Later, as the estimated position of the target, CIFI provides a greedy method for calculating the weighted average of reference points. The trained CNN model will return the most reference points, and weighted k-nearest neighbors (WKNN) will estimate the position [33]. CIFI uses fixed CNN model output, resulting in reduced positioning accuracy. For the CIFI problem, Hou et al. proposed using the Jenks natural breaks algorithm (JNBA) for adaptive selection of CNN model output [34]. This effectively solves the drawbacks of traditional fixed CNN model output. However, the JNBA has high time complexity. For the above problems, the density-based spatial clustering of applications with noise (DBSCAN) algorithm is used to select different K values adaptively, and the experiment proved that the localization effect is greatly improved [35]. DBSCAN can adaptively select a CNN output with low time complexity. The detailed procedure is shown in Section 3.

The main contributions of this paper are as follows:RSS and AOA fingerprint information are fused into RGB images as the input of the CNN model to improve the stability of the positioning system. Compared with one-dimensional input data, the CNN model can learn more information from two-dimensional input data.The K value has an important influence on the final position estimation, and the appropriate K value can improve the positioning accuracy. Compared with the traditional WKNN algorithm, the DBSCAN algorithm can choose appropriate different values of K for different situations. To a certain extent, the influence of the K value on positioning accuracy is avoided.There is a certain relationship between the input data size of the CNN model and the number of ap access points. The number of ap access points may vary according to scenarios. Since the number of ap access points and sample times are the same, we propose that the process of constructing CNN model input data is also applicable. This greatly simplifies the process of CNN model construction.

Section 4 shows the experimental setting and parameters. Section 5 shows the experimental results. Section 6 is the conclusion of this paper.

## 2. System Description

The model’s description and related parameters in the algorithm are shown in Table 1.

For this model, some common symbols should be explained. If *Q* reference points and *B* Access points exist, the experimental environment is partitioned into *Q* square regions with sides of length *l × l*. The number of *Q* depends on the length of *l*. When the length of *l* is 4, 2, or 5 m, the number of *Q* is 50, 200, or 32, respectively. The center of each square area is the label of its CNN model training set, and the reference point is located at the center of the square space [34]. Then, the vector produced by the RSS value of the *B* APs received at any point along the *i*-th label are denoted as:(1)$$r_{i}=\left(r_{1},r_{2},\cdots r_{B}\right),i=1,2,\cdots ,Q$$

The formation of the AOA vector is the same as that of RSS.
(2)$$a_{i}=\left(a_{1},a_{2},\cdots a_{B}\right),i=1,2,\cdots ,Q$$

If *T* samples are collected at a specific position on the *i*-th label, the RSS and AOA matrices generated at that time can be expressed as:(3)$$R_{i}={\left(\begin{array}{c} r_{1,1},r_{1,2},\cdots r_{1,B}\\ r_{2,1},r_{2,2},\cdots r_{2,B}\\ \vdots \cdots \cdots \cdots \cdots \vdots \\ r_{T,1},r_{T,2},\cdots r_{T,B} \end{array}\right)}_{T\times B}$$
(4)$$A_{i}={\left(\begin{array}{c} a_{1,1},a_{1,2},\cdots a_{1,B}\\ a_{2,1},a_{2,2},\cdots a_{2,B}\\ \vdots \cdots \cdots \cdots \cdots \vdots \\ a_{T,1},a_{T,2},\cdots a_{T,B} \end{array}\right)}_{T\times B}$$
where *T* is sampling frequency and *B* is the number of APs. These two matrices can be used to generate grayscale images, which are shown in Figure 1 and Figure 2.

When the above steps are followed to obtain the grayscale images of the RSS and signal arrival angle, respectively, they will be fused into an RGB image [28].

CNN fingerprint localization, like traditional indoor fingerprint localization, is divided into two stages: offline and online. Two stages of specifically related processes are shown in Figure 3.

In the offline phase, the RSS and AOA grayscales of each point in each label are fused into RGB images. To obtain the trained CNN model, the RGB pictures of label are separated into a training set and a test set. In the online stage, the RSS and AOA data gathered by target users are converted into corresponding grayscale images, and then RGB images are obtained. Then, RGB images are put into the trained CNN model for classification. Through the CNN model, the probability output of each label can be obtained. The more the CNN model predicts the output probability of the target user, the more likely the target user is to be in the region with the maximum label of the probability. To calculate the final positioning result, we need to give different K values according to different target users, and we select different K values via DBSCAN. Finally, the final estimated position is calculated by WKNN.

## 3. Description of the Positioning Algorithm

This section mainly introduces how to construct the RGB images and the corresponding labeled dataset. The section introduces several key elements of CNN model training in the offline phase. Afterwards, it introduces the selection method for different K values used by us.

### 3.1. RGB Image

If the RSS value in (Equation 3) is less than −110 dBm, it is represented by 0 [36]. Additionally, if the angle is greater than 180 degrees or less than −180 degrees, it will be represented as 180 degrees or −180 degrees, respectively. The input of CNN is generally square matrix, which requires the same number of APs and sample times. In this paper, the number of APs is ten, and the number of samples *T* should also be ten. To merge Figure 1 and Figure 2 into RGB images, one of the three channels will be set as zero. Therefore, the RGB image can be shown in Figure 4.

Before training the model, it is necessary to prepare the dataset of corresponding labels, and take *Y* points in each label. The dataset consists of Q×Y RGB pictures in total. To train the CNN model, 80% of the data were used as the training set, and the rest were used as the test set to test the model’s accuracy. Usually, the CNN model’s input is typically normalized between −1/2 and 1/2.

### 3.2. CNN Offline Training

The CNN model is made up of three layers: a convolutional layer, a pooling layer, and a fully connected layer. The convolutional and pooling layers are the two cores of the CNN. The main function of the convolutional layer is feature extraction, and the function of the pooling layer is to reduce the dimensions. Since the convolution layer and pooling layer can be randomly combined, the CNN models are also different. Classic network models include LeNet5, AlexNet, VGGNet, Google Inception Net, and ResNet. The above models increase in network depth and complexity. As the input RGB image’s size is small, LeNet5 [37] was selected as the CNN network model. Figure 5 depicts the CNN model utilized in this work.

Model learning and comprehending nonlinear functions rely heavily on the activation function. The activation function can introduce nonlinear factors to help the network learn complex problems. The activation function can map the current feature to another space. Compared with sigmoid and tanh, Relu can converge quickly in stochastic gradient descent (SGD) and is simpler to implement. Last but not least, it solves the vanishing gradient problem [38]. Therefore, the rectified linear units (Re*LU*) can be expressed as follows:(5)ReLU(x)=x,x≥00,x<0=max(0,x)

The CNN model is prone to overfitting in the training process. The overfitting manifests in the prediction probability of the model being higher and the loss function being lower in the training setcompared to the test set. Dropout can effectively solve the overfitting problem. After the first completely connected layer, 0.5 dropout is employed to improve the network model’s structure [39]. Figure 5 indicates that there are 64×3×3 features after the second pooling layer. The first fully linked layer contains 576 neurons, and the second fully connected layer contains 256 neurons. Finally, in the output layer, the probability of the *Q*-dimensional vector corresponding to *Q* labels is output. To strengthen the model, we chose cross-entropy as the loss function and selected the SGD optimizer [40].

The training CNN model can find the nonlinear mapping relationship between the test set and its corresponding label. With the increase in training times, the loss function gradually decreases and tends to be stable. At this time, the CNN model is almost finished with training. The relevant parameters of the network model will be saved.

### 3.3. Improved DBSCAN Algorithm

#### 3.3.1. The Reason for Using DBSCAN to Select K

In general, the traditional method is to choose a fixed value of K to calculate the error. However, this method cannot choose different K values according to different situations.

In theory, if the model of CNN classification output is reliable enough, an ideal K value can considerably enhance positioning accuracy. For example, when k is equal to 1, the target user will fall in a reference point grid. If the maximum probability output of the model prediction is correct, the error will not show too much deviation. If the model prediction is wrong, the error is bound to increase. In this case, when K is equal to 2 or 3, the maximum output probability of the model is most likely wrong, and neighboring reference points have to be added. When K is equal to 2, the first two with the largest output probability of the CNN model, namely, the two points of reference nearest to the target user, are selected for position estimation. When K is equal to 3, the first three with the largest output probability of the CNN model, namely, the three points of reference nearest to the target user, are selected for position estimation. This observation also reflects the similarity between the reference points selected by the above K value. There is an adjacent relationship between them in the actual position. The most intuitive expression of such similarity in CNN output is that the output probability is similar or even the same. As we all know, the CNN model output will return the probability corresponding to each label, and each label corresponds to each reference point. When the corresponding RGB image of an unknown user is put into the trained CNN model, the reference point with a high output probability of the CNN model is likely to be the location of the target user. Combining CNN with indoor positioning is a basic classification problem.

DBSCAN is a clustering classification algorithm that can categorize CNN output to acquire an acceptable K value and relevant reference points.

#### 3.3.2. The Process of the Improved DBSCAN Algorithm Selecting a K Value

DBSCAN is a spatial clustering algorithm based on density. Its essence is to discover high-density datasets in a dataset. These high-density data are divided into different clusters. Suitable clusters are selected as references for K-value selection [41].

The algorithm uses epsilon, minPts, and the core point parameters to determine the threshold of the high-density dataset. This algorithm can divide the output of the CNN model into several different clusters. Epsilon refers to the distance between an object J and objects O. $N_{epsilon}\left(J\right)$ contains all objects in dataset *D* that are no more than epsilon away from object *J*, which is:(6)$$N_{epsilon}\left(J\right)=\left\{o\in D|Dist(j,o)\leq epsilon\right\}$$
where *D*, *Dist*(*j*,*o*) and $N_{epsilon}\left(J\right)$ denote the dataset, the distance between object *J* and object O and that between object *J* and every object in dataset *D* whose distance from object *J* is not greater than epsilon, respectively. MinPts is the minimum of points in the $N_{epsilon}\left(J\right)$. In addition to the above two important parameters, there is another important parameter core point. Given dataset *D*, neighborhood density threshold minPts is set. If there is an object j∈D and Formula (Equation 7) is satisfied, then J is a core center.
(7)$$\left|N_{epsilon}\left(j\right)\right|\geq minPts$$

With the three important parameters above, different parameter settings will obtain different classification results. After many attempts, when epsilon = 2, minPts = 1, the classification works better, and similar reference points will be grouped into one class.

To demonstrate how DBSCAN works, the following example is shown. The center point is the same. However, the three data points around it are not quite the same. If the data are not the same, the amount of data in each cluster is not the same. To reduce the calculation burden, we selected 1 for both epsilon and minPts. We calculated the distance from each data point to the center point. If the distance was less than the epsion, this point and the center point were grouped into a cluster. Table 2 displays all of the classifications.

### 3.4. Online Localization

#### 3.4.1. Different Choices of K

First of all, the output value of the CNN model online positioning was converted from one-dimensional to two-dimensional and sorted in descending order. Secondly, the largest model output value was selected as the core object of DBSCAN. In this paper, epsilon as 2 and minPts as 1 were selected for classification. Through the above steps, we selected the classification result centered on the core object, and the general classification result was not to exceed 3. The details are shown in Algorithm 1.
**Algorithm 1** Different DBSCAN-based K value selections.**Input** **:***P* from CNNs, $P\in R^{1\times Q}$, epsilon, minPts, D.1:Sort *P* in descending order of size, and go from one dimension to two dimensions;2:$P_{max}\leftarrow P$, epsilon = 2, minPts = 1, D = 0, K = 0;3:**while** j=1 to *Q* **do**4:    $D=\sqrt{{\left[p_{\max}\left(x\right)-p_{j}\left(x\right)\right]}^{2}+{\left[p_{\max}\left(y\right)-p_{j}\left(y\right)\right]}^{2}}$5:    **if** D<epsilon **then**6:        K++;7:    **else**8:        *K*;9:    **end if**10:**end while****Output** **:***K*, K∈(1,2,3)

A concrete example in Table 3 displays how Algorithm 1 performs. The output of CNN for fifty reference points is *P* = (−1, 3, 5, 3, 7, −1, 1, −2, −1, 0, −4, −5, −2, −4, −1, 0, 0, 0, 12, 9, −4, −7, −5, −3, −6, −5, −5, −1, −3, 1, −5, −1, −2, 0, 3, −4, 2, 2, 0, 0, −1, 1, 3, 5, 11, 13). The data were converted from one-dimensional to two-dimensional.

The sequential output probabilities of CNN model are ‘10-10’, ‘10-14’, ‘10-18’, ‘10-02’, ‘10-06’, ‘14-10’, ‘14-14’, ‘14-18’, ‘14-02’, ‘14-06’, ‘18-10’, ‘18-14’, ‘18-18’, ‘18-02’, ‘18-06’, ‘02-10’, ‘02-14’, ‘02-18’, ‘02-02’, ‘02-06’, ‘22-10’, ‘22-14’, ‘22-18’, ‘22-02’, ‘22-06’, ‘26-10’, ‘26-14’, ‘26-18’, ‘26-02’, ‘26-06’, ‘30-10’, ‘30-14’, ‘30-18’, ‘30-02’, ‘30-06’, ‘34-10’, ‘34-14’, ‘34-18’, ‘34-02’, ‘34-06’, ‘38-10’, ‘38-14’, ‘38-18’, ‘38-02’, ‘38-06’, ‘06-10’, ‘06-14’, ‘06-18’, ‘06-02’ and ’06-06’ corresponding to labels from 1 to 50. The label corresponding to the CNN model that selects the K value needs to be defined. In Section 2, the label has been defined. After the above data are processed by Algorithm 1, (1,13) and (1,12) are divided into the same cluster.

#### 3.4.2. Location Estimation

When the output of the CNN model is processed, each result contains K largest outputs. Of course, K is not fixed at each target point. Suppose that when K is equal to 1, the maximum CNN output value is not comparable to other values. Additionally, the estimated position of the final target user is the reference point position corresponding to the maximum value. If K is greater than one and the K biggest outputs share some features, the estimated locations are determined using their respective weights, as shown below:(8)$$\hat{L}=\sum_{i=1}^{K}l_{index(i)}\times \frac{P_{out}\left(i\right)}{\sum {}_{j=1}^{K}P_{out}\left(j\right)}$$
where $l_{i}$ is the reference point corresponding to the *i*-th maximum probabilities of the CNN model output, and $P_{out}$ is the first K maximum output probability.

## 4. The Experimental Setting and Parameters

The hardware and software configurations in the experiment were as shown in Table 4.

The experimental region was divided into fifty 4 m × 4 m grids, and the suitable reference points were determined.

The simulation scene was based on No.2 building of Nanjing University of Information Science and Technology, and the experimental area was 40 m × 20 m. The experimental region was separated into a large number of 4 m × 4 m grids, and the suitable reference points were determined.

Figure 6 represents a schematic design of the simulation area’s architecture. Since the spacing of reference points was 4 m, our final number of reference points was 50. The locations of the reference points within each labeled area were determined, and a data point was taken at 0.18 m intervals, resulting in 529 data points in a label. The RSS and angle information were collected by Wireless InSite software. Related parameters needed to be set. The transmitting antenna of each AP had a sinusoidal waveform and a carrier frequency of 2.4 GHz [42]. The transmitting power of the signal was set to 150 dBm. There were a total of 26,450 data used for CNN training and testing. There were 5290 data for online validation.

In this paper, the proposed method is compared with the traditional CNN method. There were three alternative fixed K values for the classic CNN approach—namely, K was 1, 2, 3, 4 or 5. However, the proposed method can select the appropriate value of K according to different situations.

All approaches employed the same dataset and CNN model to ensure the consistency of experimental outcomes. The detailed parameters of CNN are shown in Table 5:

## 5. Numerical Results

Figure 7 shows the cumulative distribution function (CDF) diagram when the distance between the reference points is 4 m. As may be seen in the CDF diagram, the steeper the curve trend, the better the positioning effect. Therefore, compared with the traditional KNN method, the positioning effect of the proposed method is better. Specifically, positioning errors of less than 1 m occurred at 40% of the test spots. However, when K was 1, 2, 3, 4 and 5, the error of traditional CNN was less than 1m, which is 19%, 31%, 37%, 34% and 35%, respectively. On the other hand, 99% of the test points generated by our proposed method had positioning errors of less than 3 m. Meanwhile, the positioning errors of other traditional methods were less than 3 m, accounting for 95%, 97%, 96%, 93% and 88%, respectively.

Figure 8 and Figure 9 are the other two control tests, and Figure 8 is the CDF diagram between the two reference points with a distance of 5 m. Similarly, compared with the reference point distance of 4 m, the Flexible K value method has a better positioning effect. In the proposed method, 73% of the test points are located less than 2 m. However, the other schemes were 47%, 65%, 65%, 60%, and 57%, respectively.

Figure 9 shows the CDF diagram when the distance between the reference points is 2 m. Compared with the above two points, the narrower the distance between reference locations, the higher the positioning accuracy and the better the effect. However, it takes extra time to build a more dense fingerprint database. In the proposed method, 77% of the test point positioning errors are less than 1m. When K is 1, 2, 3, 4, 5, the error of traditional CNN is less than 1 m, which is 19%, 31%, 37%, 66%, 67%, respectively. From the above experiments, it can be seen that the choice of K value has a significant impact on the calculation of the final position. In short, high positioning accuracy can be obtained by adaptive selection of appropriate K value.

Figure 10 shows the comparison of the average positioning error between the traditional algorithm and the proposed method when the distances between the reference points are different. Within a certain distance from the reference point, the average positioning error decreases as the distance between reference sites decreases, and the positioning effect increases.

When the distance between reference points is small, the divided square area is denser. However, it takes more time to collect fingerprint database information in the offline phase. On the other hand, when the distance between reference points is the same, the average error of our proposed method is the lowest, and the positioning effect is better.

On the whole, when the distance between reference points is small, we need to spend more time collecting data to build the fingerprint database. At the same time, the number of labels required in the CNN model will increase. Importantly, the positioning accuracy will be significantly improved. When the distance between reference points is large, we do not need to spend much time to collect data. The labels needed in the CNN model will also be reduced. At the same time, the positioning accuracy will be reduced.

## 6. Conclusions

In this paper, the DBSCAN algorithm was used to select the output of a CNN model adaptively. This algorithm solves the problem of reference-point selection for unknown locations in CNN model output prediction. We also propose to construct RGB images by fusing RSS and AOA as the input of CNN model. This fusion method greatly increases the stability and robustness of CNN model. A large number of simulation results showed that the localization accuracy of our proposed method is improved compared with the traditional CNN localization estimation method. The positioning accuracy will be improved by at least 0.1 to 0.3 m. Although our research has played a role in increasing the positioning accuracy, it still faces many challenges. In a relatively stable environment, if a Wi-Fi device fails suddenly or the Wi-Fi signal fluctuates frequently, the collection of data-source information will be affected, which will certainly affect our final positioning estimation. In future research, we intend to predict and locate the dynamic motion trajectory. We need to design a local space-feature-extraction scheme based on a CNN. Secondly, the depth LSTM model should be introduced to establish the correlation between consecutive frames. In addition, we can add other information on the basis of two-dimensional information, so that our input information will be changed from two-dimensional to three-dimensional.

## Figures and Tables

**Figure 1 sensors-22-09531-f001:**
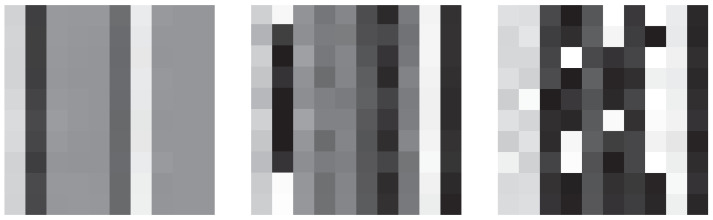
Received signal strength (RSS) grayscale under different labels (Labels from left to right are ‘2-2’, ‘2-6’, and ‘2-10’ respectively).

**Figure 2 sensors-22-09531-f002:**
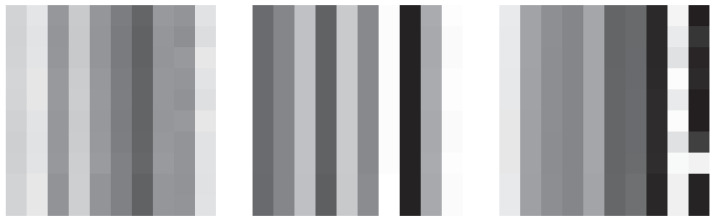
Angle of arrival (AOA) grayscale under different labels (Labels from left to right are ‘2-2’, ‘2-6’ and ‘2-10’ respectively).

**Figure 3 sensors-22-09531-f003:**
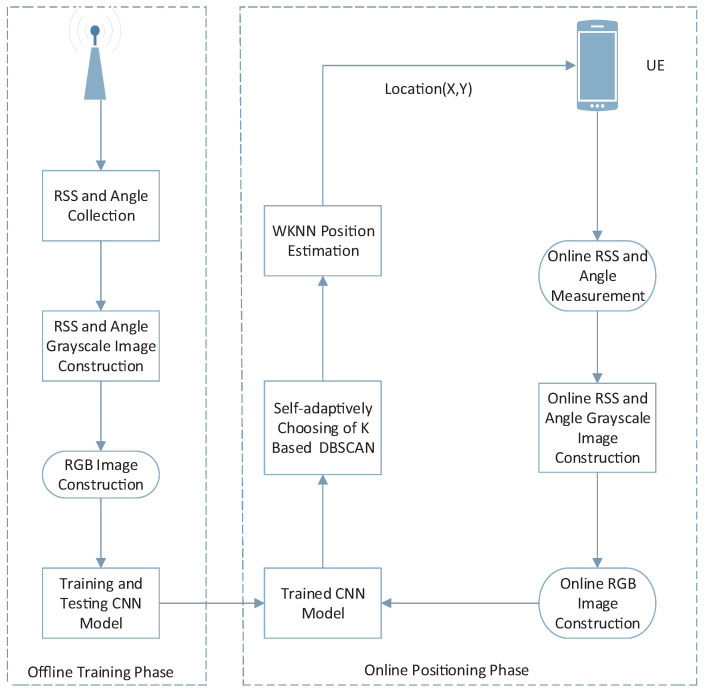
Complete CNN positioning system implementation (offline training phase and online positioning phase).

**Figure 4 sensors-22-09531-f004:**
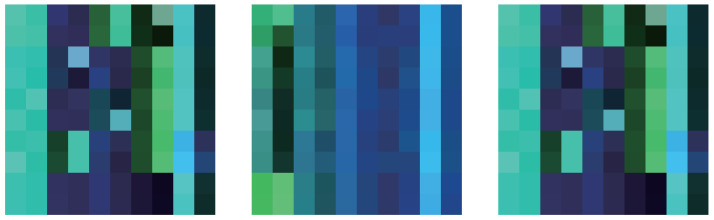
RGB images under different labels (Labels from left to right are ‘2-2’, ‘2-6’ and ‘2-10’ respectively).

**Figure 5 sensors-22-09531-f005:**
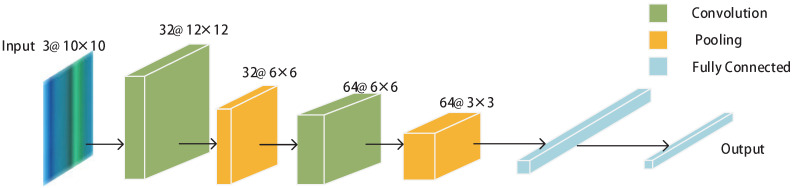
System design of the LeNet5 CNN network model.

**Figure 6 sensors-22-09531-f006:**
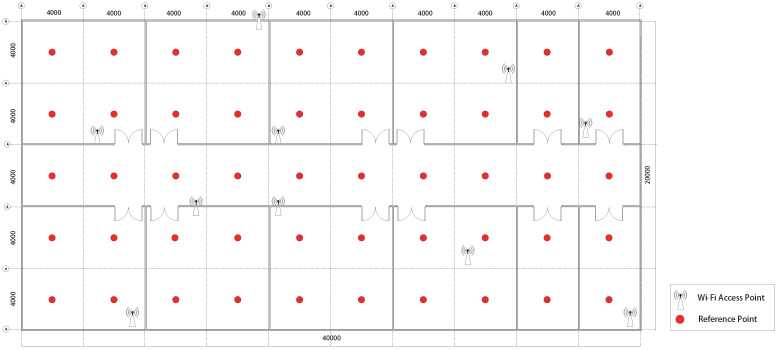
Setting arrangement of simulation area when the number of reference points is 50.

**Figure 7 sensors-22-09531-f007:**
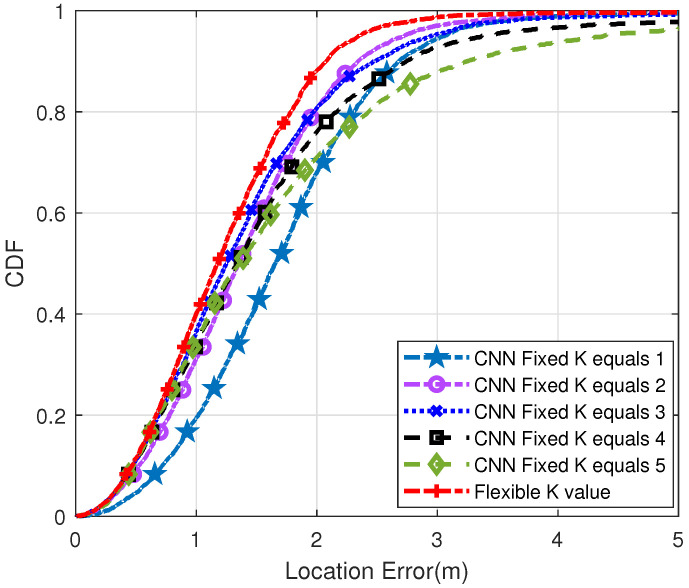
Probability distribution diagram of positioning error when the distance between two reference points is 4 m (Q = 50).

**Figure 8 sensors-22-09531-f008:**
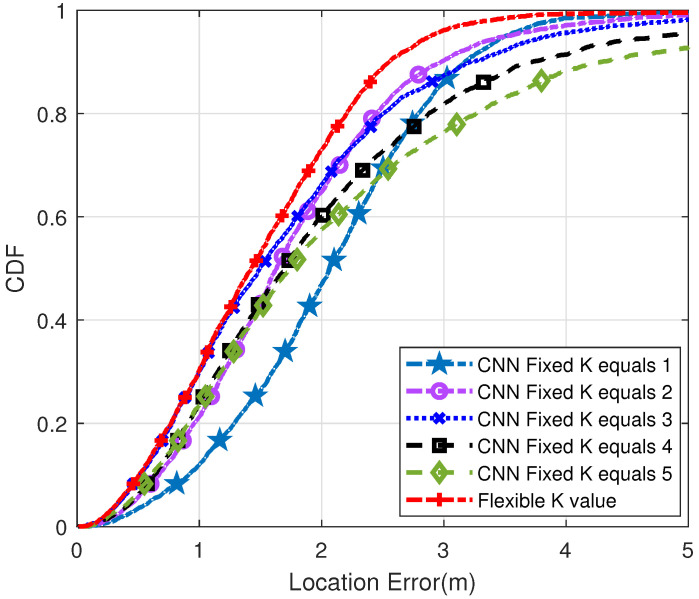
Probability distribution diagram of positioning error when the distance between two reference points is 5 m (Q = 32).

**Figure 9 sensors-22-09531-f009:**
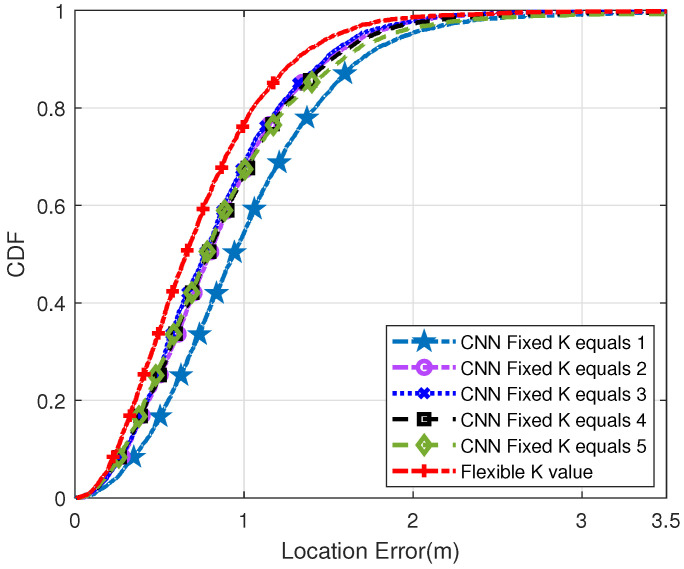
Probability distribution diagram of positioning error when the distance between two reference points is 2 m (Q = 200).

**Figure 10 sensors-22-09531-f010:**
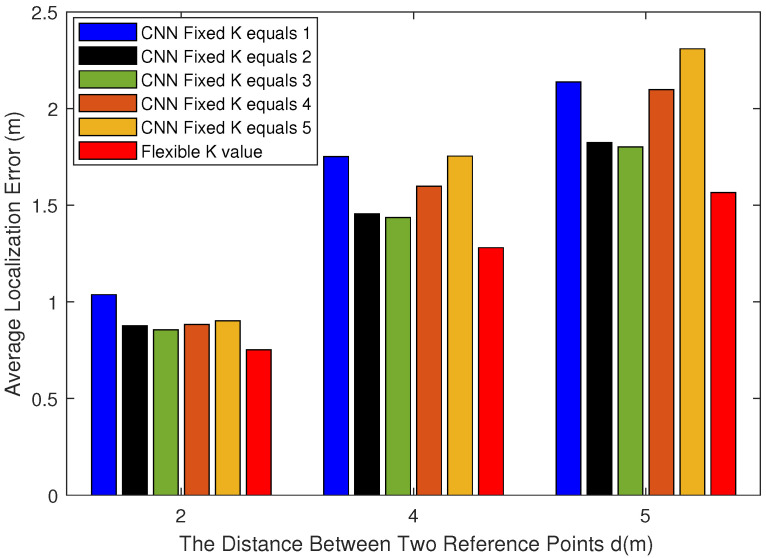
The average localization error for different reference point distances.

**Table 1 sensors-22-09531-t001:** Model and algorithm-related symbols.

Notation	Definition
*Q*	The Number of Reference Points
*B*	The Number of Access Points
*l*	The Side Length of square
$r_{i}$	The RSS Vector
$a_{i}$	The AOA Vector
*T*	The Number of Samples
*Y*	The Number of Sampling Points in Each Label
*D*	The Distance between center point and data point
*P*	The Probability of CNN Model Output Label

**Table 2 sensors-22-09531-t002:** DBSCAN classification example result display.

The Center Point	Data_1	Data_2	Data_3	Classification
(1,1)	(0,0)	(2,1)	(3,2)	(1,1)(2,1)
(1,1)	(0,1)	(2,1)	(3,2)	(1,1)(0,1)(2,1)
(1,1)	(0,2)	(2,3)	(3,2)	(1,1)

**Table 3 sensors-22-09531-t003:** The output of the real CNN model is selected by Algorithm 1.

The Core Point	Two-Dimension Data	Classification
(1,13)	(1,13),(2,12),(3,11) (4,9),(5,7),(6,5) (7,5),(8,3),(9,3) ⋯⋯⋯ (48,−5),(49,−6),(50,−7)	(1,13) (2,12)

**Table 4 sensors-22-09531-t004:** Software and hardware used in simulation experiment and corresponding version numbers.

Hardware/Software	Model/Version Number
CPU	AMD Ryzen 5 5600 H
GPU	Nvidia GTX 1650
RAM	16 G
MATLAB	R2021a
Python	3.6
Wireless Insite	3.3.5

**Table 5 sensors-22-09531-t005:** Specific relevant parameters of each layer in the CNNs.

CNN Structure	Parameters
Input Size	3@10 × 10
Convolution 1	32@3 × 3, Stride:1, Padding:2
Convolution 2	64@3 × 3, Stride:1, Padding:1
Activation Function	ReLU
Pooling Layers	2 × 2 MaxPool
Fully Connected	63 × 3 × 3 to 256
Output	256 to *Q*

## Data Availability

Not applicable.

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
