# Peer review of "Improved CNN-Based Indoor Localization by Using RGB Images and DBSCAN Algorithm"

_sensors, 2022, doi:10.3390/s22239531_

Round 1
Reviewer 1 Report
The presented manuscript proposes a position estimation method based on DBSCAN algorithm for appropriate reference points selection. DBSCAN analyses the output from the Convolutional Neural Network and chooses the number of reference points, and the weighted K-nearest neighbors eventually approximate the position. Moreover, the RSS and AOA fingerprint information are fused into RGB images as the input of Convolutional Neural Network to improve the stability of the positioning system.
The manuscript is well written, it has a good literature review, the organization is good, and the method development as well as the results are clearly presented. However, it still needs to be improved.
My comments in more details are as follows:
-It seems like Lines 91-99 of the Introduction section provide an abstract of the methodology. It is suggested to replace this part with a brief statement regarding the novelty of the research as well the main objectives of the conducted study. The methodology of the study is fully expanded in the following sections and there is no need for it to be explained in the Introduction section.
-Line 100 of the Introduction section reads: ” The main contributions of this paper are as follows”, while the following sections are more like information regarding the the model development procedure. The contributions of the study should explain the benefits and importance of the findings for the defined problem.
-The current Introduction section in composed of many paragraphs, which somehow miss the flow of the information towards the necessity and objective of the research. I think it is better to merge some paragraphs and create more connection between its different parts.
-Captions of some figures and tables should be more informative. For example, the current captions for the Figure 3 and Table 3 are short and can not be individually helpful.
- All the parameters related to model adjustment mentioned for the method development should be included in a table. In other words, a Table should be created to include all the parameters mentioned in the text from section 2 to 4.
-The information provided in section 3 (Description of the positioning algorithm) mostly repeat the currently available methods and do not provide new information related to this research. For example, major parts of the section 3.2 (CNN Offline Training) read the conventional CNN algorithm and its different parts without getting much into its link with the research objective. More strengthening of the method description in relation to the research objective is required.
-I think It is better to move the current results mentioned in section 3.3 to the Result’s section in section 4. For example, Table 1 can be expanded and discussed with a better focus in the related Result’s section.
-In the current format, the experimental setting is merged with Results of the study. I suggest to separate experimental setting and bring it before starting the Results. The Results section should be devoted to presentation and comparison of the results of the methods already explained in the previous methodology parts.
- Some words in the texts start with Capital letter in middle of the sentences. To mention some: line 21: Intensity, line 27 WI-FI (should change to Wi-Fi), line 30: Since, line 34 Angle. Please correct them.
Reviewer 2 Report
MDPI Sensors Journal (Manuscript ID: sensors- 2037823)
Comments to the Author
This paper investigates a position estimation method using RGB images via DBSCAN algorithm. It is a useful topic and the paper studies the technical concepts clearly. However, there are several points need to be addressed to improve the quality of the manuscript.
Suggestions to improve the quality of the paper are provided below:
1) Structural issues in the abstract:
a. Certain acronyms "AOA", "RGB" and ”DBSCAN” are not defined at the beginning when they are first used in the abstract. Please double check that all acronyms are properly defined when they are first used.
2) Clarity of the abstract:
a. Abstract should be written more clearly and concise. Currently, the short description of the methodology in the abstract, sounds very disjoint. For example, it is mentioned about “Density-Based Spatial Clustering of Applications” but it does not connect well with the title or “improved CNN”. How does improved CNN play a role in the proposed approach needs to explained. Please structure the abstract in a more reader friendly form. (Line 11-14)
3) Wireless technologies have been widely used in indoor positioning systems. Authors only mentioning about range-free and range-based Wi-Fi indoor positioning technologies (Line 27), whereas an overview should also be given other impactful wireless technologies such as Bluetooth and RFID for indoor positioning being inexpensive and widely available. Please include a short overview of the suggested literature and provide a reasoning of the selected wireless technology (Wi-Fi).
Filippoupolitis, A., Oliff, W. and Loukas, G., 2016, October. Occupancy detection for building emergency management using BLE beacons. In International Symposium on Computer and Information Sciences (pp. 233-240). Springer, Cham.
Huang, Ke, Ke He, and Xuecheng Du. "A hybrid method to improve the BLE-based indoor positioning in a dense bluetooth environment." Sensors 19.2 (2019): 424.
Tekler, Z.D., Low, R., Gunay, B., Andersen, R.K. and Blessing, L., 2020. A scalable Bluetooth Low Energy approach to identify occupancy patterns and profiles in office spaces. Building and Environment, 171, p.106681.
Hahnel, Dirk, et al. "Mapping and localization with RFID technology." IEEE International Conference on Robotics and Automation, 2004. Proceedings. ICRA'04. 2004. Vol. 1. IEEE, 2004.
Li, N. and Becerik-Gerber, B., 2011. Performance-based evaluation of RFID-based indoor location sensing solutions for the built environment. Advanced Engineering Informatics, 25(3), pp.535-546.
4) The authors should clearly state the novelty of the paper, which is currently missing from the manuscript. Although, the main contributions have been listed clearly, authors should more clearly highlight the novelty of the proposed approach or how it improves upon the existing literature and not just a description of what was done. For instance, the authors mentioned that the DBSCAN algorithm was introduced to “select the appropriate reference points according to the situation”, but what are the advantages of that?
5) I highly suggest authors to mention about the applications that leverage indoor localisation technologies to further emphasize the importance of the field studied. Especially in the building domain, there are many applications of indoor positioning systems have been used such as building emergency management, smart plug and HVAC controls. Please refer to the following readings.
Indoor positioning for building emergency management
Filippoupolitis, A., Oliff, W., & Loukas, G. (2016, December). Bluetooth low energy based occupancy detection for emergency management. In 2016 15th international conference on ubiquitous computing and communications and 2016 International Symposium on Cyberspace and Security (IUCC-CSS) (pp. 31-38). IEEE.
Indoor positioning for smart plug load control
Tekler, Z.D., Low, R., Yuen, C. and Blessing, L., 2022. Plug-Mate: An IoT-based occupancy-driven plug load management system in smart buildings. Building and Environment, 223, p.109472.
Indoor positioning for smart HVAC controls
Balaji, B., Xu, J., Nwokafor, A., Gupta, R. and Agarwal, Y., 2013, November. Sentinel: occupancy based HVAC actuation using existing WiFi infrastructure within commercial buildings. In Proceedings of the 11th ACM Conference on Embedded Networked Sensor Systems (pp. 1-14).
6) Apart from DBSCAN, there are many other clustering algorithms such as K-means++, BIRCH, and Spectral Clustering. Why did the authors decided to only use DBSCAN?
7) The authors should provide a short discussion about the choice of Q (i.e., the number of reference points) in the Experimental Settings section. I imagine that a large Q would result in a more high-resolution model but has a negative impact on the model’s performance, vice versa. Did the authors consider other Q values?
8) In Figure 7, 8, and 9, the authors have only shown the results when K is fixed at 1, 2 and 3. However, the performance seems to generally improve when K increases. I suggest that the authors include the results for K > 3 as well so that we can see if the model performance continues to improve or if it will start to fall after some time. In that case, this will provide an even stronger justification for a flexible K value approach.
9) Limitations of the proposed approach should be highlighted.
10) Finally, I suggest the authors to discuss and expend the conclusion about potential future directions that can be explored using proposed approach and what other improvements can be made to the proposed framework in future works.
Round 2
Reviewer 1 Report
The authors have addressed my comments and implemented my suggestion within the manuscript.
However, the current title which was changed during the revision can be improved. I suggest that the authors improve the title before publication.
Author Response
Thank you very much for your suggestions on the revision of the title. I have revised it again in the new manuscript and highlighted it.

Reviewer 2 Report
Thank you for addressing my comments and concerns. The current version of the manuscript is clearer and more comprehensive. Great job!
Author Response
Thank you for your hard work and valuable suggestions. Thank you very much.